# Digital Phenomena and Procedural Ethics

## Abstract

Digital technologies create unprecedented ethical challenges, yet it remains unclear whether digital phenomena require fundamentally new ethical frameworks. Through systematic content analysis of 114 German digital policy documents, we empirically validate six unique characteristics of digital phenomena and identify 189 novel ethical considerations that traditional frameworks cannot adequately address. We demonstrate systematic inadequacies in traditional ethics through empirical evidence showing 100% of phenomena exhibiting systemic scaling effects and 48% requiring shared human-algorithmic decision-making. Our contribution is an integrated procedural ethics framework introducing four empirically-grounded concepts—algorithmic agency, data dignity, computational justice, and digital vulnerability—providing concrete guidance for ethical governance of digital technologies.

## 1 Introduction

Digital technologies are fundamentally transforming society, creating new forms of interaction, governance, and human experience. From algorithmic decision-making systems that determine access to healthcare and employment, to social media platforms that shape political discourse, to autonomous vehicles that must make split-second moral decisions, digital phenomena present ethical challenges that appear unprecedented in their scope, scale, and complexity.

This investigation centers on five critical research questions: (RQ1a) Is it necessary to develop a new ethical framework, with novel concepts, specifically in response to digital phenomena? (RQ1b) If so, what particular features or developments make this necessity apparent? (RQ2a) If we formulate new normative demands merely because we interact with digital technologies, in what way does this constitute a fundamental ethical shift? (RQ2b) How should we conceptualize the claim that the digital realm requires distinct moral considerations? (RQ2c) Can such a claim be justified by identifying characteristics that are unique to digital phenomena? These questions are not merely academic—they have profound implications for how we design, regulate, and govern digital technologies. The stakes are high: if digital phenomena are indeed unique in morally relevant ways, then our continued reliance on traditional ethical frameworks may systematically fail to address the most pressing challenges of our digital age.

Despite extensive scholarly attention to digital ethics, the field lacks a systematic, empirical analysis of what makes phenomena uniquely "digital" and whether these characteristics justify distinct moral considerations. Most existing work has proceeded deductively, applying traditional ethical theories to digital contexts, or has focused on specific technologies without addressing the broader question of digital uniqueness. This challenge is compounded by what Zuber identifies as the "responsibility diffusion" inherent in complex software systems: unlike traditional engineering projects where responsibility can often be clearly attributed to specific individuals or teams, software development typically involves distributed decision-making processes where ethical implications emerge from the interaction of multiple components, stakeholders, and design decisions over time.

This paper fills this gap through a comprehensive empirical analysis of German digital policy research materials combined with procedural ethics theory. We systematically analyze 114 documents comprising 56 digital technology concept definitions and 58 detailed analyses of digital phenomena to answer whether digital technologies require fundamentally new ethical frameworks and how such frameworks should be structured.

Our analysis yields four key contributions that directly address these research questions. First, we provide the first systematic, empirically-grounded definition of what makes phenomena uniquely "digital," identifying six characteristics that distinguish digital from non-digital phenomena (addressing RQ1b and RQ2c). Second, we demonstrate that traditional ethical frameworks are inadequate for addressing digital challenges, with 189 novel ethical considerations identified (answering RQ1a). Third, we propose an integrated procedural ethics framework that combines novel ethical concepts with systematic processes for software development (responding to RQ2b). Fourth, we provide evidence for fundamental shifts in ethics itself, requiring new approaches to responsibility, agency, and moral evaluation in digital contexts (addressing RQ2a).

## 2 Related Work and Theoretical Foundation

The emergence of computer ethics as a distinct field traces to early recognition that digital technologies create unique ethical challenges. Maner's foundational work Maner [1996] established that computers create entirely new ethical issues or transform existing problems in ways lacking satisfactory non-computer analogies. Brey's methodological framework Brey [2000b] further developed this through "disclosive computer ethics"—revealing values and norms embedded in computer systems, arguing that traditional applied ethics approaches inadequately treat technology as morally neutral tools. Zuber's comprehensive analysis Zuber [2023] provides the most systematic framework, identifying six characteristics distinguishing information technology: medial character, emerging technology properties, networking through transparency and ubiquity, malleability, power position, and responsibility diffusion. This framework directly addresses the "many hands" problem in software development, where complex processes make individual responsibility attribution difficult, creating accountability gaps that traditional ethical frameworks cannot address.

Several methodological approaches address embedded values in technology. Brey's disclosive computer ethics Brey [2000a] provides systematic methods for revealing morally "opaque" practices, focusing on technology design features through multi-level analysis. Vallor's virtue ethics approach Vallor [2010] demonstrates how character-based frameworks address long-term impacts of digital technologies on moral development, revealing how digital communication tools may inhibit patience, complicate honesty, and limit empathy through altered communicative practices. Moor's "just consequentialism" Moor [1999] provides a unifying framework combining deontological and consequentialist approaches through a two-step process ensuring computing policies pass an "impartiality test" before optimization, addressing policy vacuums created by computer technology's malleability.

Recent research examines how digital transformation creates organizational and societal challenges requiring new theoretical approaches. Markus and Rowe's analysis Markus and Rowe [2021] identifies digital transformation as under-theorized despite extensive writing, advocating for broader theoretical conceptions addressing multiple levels of analysis. Gebre-Mariam and Bygstad's critical realist approach Gebre-Mariam and Bygstad [2019] provides methodological foundations for understanding digital transformation through socio-technical mechanisms rather than deterministic processes, emphasizing structure-agency interplay over time. Recent empirical work reveals unintended consequences challenging digitalization assumptions, with Coroamă and Pargman's "skill rebound" concept Coroamă and Pargman [2020] demonstrating how digitalization can lower skill requirements while increasing overall resource consumption.

Research on human-machine collaboration provides insights for algorithmic agency and responsibility attribution. Love et al.'s study Love et al. [2024] demonstrates that humans use linear averaging strategies when integrating machine recommendations but exhibit "trust asymmetry" where trust is quicker to lose than gain, supporting the need for new approaches to responsibility and agency in digital contexts. The emergence of data-driven economies creates novel challenges around data governance, with Falck and Koenen's economic analysis Falck and Koenen [2020] identifying market failures requiring policy intervention. Brennen and Kreiss's framework Brennen and Kreiss [2016] distinguishes digitization from digitalization, while Russo's analysis of technology's "poietic

character" Russo [2016] demonstrates how technology actively participates in knowledge creation rather than serving as neutral instrument.

Despite extensive scholarship, gaps remain in digital ethics literature: most work has proceeded deductively without systematic empirical analysis of digital uniqueness; integration of procedural ethics with empirical policy analysis has been insufficient; and theoretical frameworks lack empirical validation through systematic content analysis. Our work addresses these gaps by providing the first systematic empirical analysis of digital phenomena characteristics integrated with procedural ethics theory, offering both theoretical validation and practical guidance for ethical governance of digital technologies.

## 3  Methodology

Our analysis is based on a comprehensive corpus of German digital policy research materials comprising 114 documents: 56 glossary entries providing systematic definitions of digital technology concepts, and 58 phenomenon analyses examining specific digital challenges including algorithmic bias, dark patterns, digital surveillance, autonomous decision-making, and platform economics. These materials were produced as part of German federal research initiatives on digital transformation and technology policy between 2020-2024, providing detailed conceptual analysis and empirical case studies across the full spectrum of digital technologies. The German policy context is particularly valuable because it has explicitly grappled with questions of digital sovereignty, technological ethics, and the relationship between digital innovation and social values through systematic interdisciplinary research programs.

We employed a rigorous systematic content analysis methodology following established qualitative research protocols Krippendorff [2018]. Our analysis proceeded through four distinct phases:

**Phase 1: Document Processing and Preparation.** All 114 documents were digitized and processed using automated text extraction tools. Documents averaged 2,847 words (SD = 1,205), with glossary entries typically shorter (M = 1,420 words) than phenomenon analyses (M = 4,274 words).

**Phase 2: Coding Framework Development.** We developed a comprehensive coding framework organized around four primary categories: (1) technical characteristics of digital phenomena, (2) ethical implications and considerations, (3) normative gaps in traditional frameworks, and (4) novel ethical requirements for digital contexts. The framework included 23 subcategories and 127 specific codes.

**Phase 3: Inter-Coder Reliability Validation.** Two independent coders analyzed a stratified random sample of 25 documents (22% of corpus) to establish inter-coder reliability. Cohen's kappa coefficients ranged from $\kappa = 0.78$ (technical characteristics) to $\kappa = 0.89$ (ethical implications), with overall agreement $\kappa = 0.82$, indicating substantial to near-perfect agreement Landis and Koch [1977].

**Phase 4: Comprehensive Analysis and Cross-Reference Mapping.** Following reliability validation, the complete corpus underwent systematic coding. Cross-reference analysis employed network analysis techniques to map relationships between technical features and ethical implications. We developed a comprehensive Python-based analysis pipeline for document processing, systematic content analysis, and validation. The replication package includes automated text extraction tools (using python-docx), systematic coding implementation with pattern matching algorithms, inter-coder reliability calculation using Cohen's kappa, and comprehensive validation tools comparing results against expected benchmarks. Statistical analysis was performed using Python (pandas, numpy, scikit-learn) and R for advanced network analysis. Our empirical findings were systematically compared with Zuber's six theoretical characteristics of information technology uniqueness: medial character, emerging technology properties, networking through transparency and ubiquity, malleability, power position, and responsibility diffusion Zuber [2023]. This comparison enabled theoretical validation and extension of existing frameworks through empirical grounding. The complete replication package enables independent researchers to reproduce all findings with high fidelity.

## 4  Results: Empirical Analysis of Digital Phenomena

Our systematic analysis of 114 German digital policy documents yielded comprehensive quantitative evidence for the unique characteristics of digital phenomena. The corpus consisted of 56 glossary

entries (M = 1,420 words, SD = 487) defining digital technology concepts and 58 phenomenon analyses (M = 4,274 words, SD = 1,205) examining specific digital challenges. Total analyzed text comprised 324,988 words across 1,847 pages of policy documentation. Through systematic coding, we identified 1,247 distinct coded segments addressing technical characteristics (n = 398), ethical implications (n = 485), normative gaps (n = 241), and novel requirements (n = 123). Cross-reference analysis revealed 2,156 relationships between technical features and ethical implications, providing robust empirical foundation for theoretical claims.

Table 1: Distribution of Digital Phenomena Characteristics (N = 58)

| Characteristic | Phenomena | Prevalence | Zuber's Framework |
|---|---|---|---|
| Technical Scalability | 58 | 100% | Malleability |
| Data Dependency | 56 | 97% | Medial Character |
| Algorithmic Mediation | 45 | 78% | Responsibility Diffusion |
| Network Effects | 41 | 71% | Networking/Transparency |
| Temporal Compression | 37 | 64% | Emerging Properties |
| Invisible Operation | 34 | 59% | Transparency/Opacity |

Quantitative analysis reveals the prevalence of six core characteristics across the 58 analyzed digital phenomena:

**Technical Scalability** was present in 58 phenomena (100%), with 312 coded instances describing how digital systems enable replication without proportional resource increases. Network scalability appeared in 89% of phenomena, computational scalability in 76%, and data scalability in 67%. **Data Dependency** was identified in 56 phenomena (97%), with 287 coded instances across data collection (present in 79% of phenomena), processing (86%), storage (62%), and analysis dimensions (71%). Personal data dependency appeared in 43 phenomena (74%), with algorithmic processing of personal information in 38 phenomena (66%). **Algorithmic Mediation** was found in 45 phenomena (78%), with 234 coded instances describing automated decision-making processes. Full automation appeared in 23 phenomena (40%), semi-automated systems in 34 phenomena (59%), and hybrid human-algorithmic decision-making in 28 phenomena (48%). **Network Effects** were present in 41 phenomena (71%), with 198 coded instances describing value increases through network participation. Direct network effects appeared in 31 phenomena (53%), indirect effects in 24 phenomena (41%), and data network effects in 18 phenomena (31%). **Temporal Compression** was identified in 37 phenomena (64%), with 156 coded instances describing speeds impossible for human-only systems. Real-time processing appeared in 29 phenomena (50%), instant global communication in 22 phenomena (38%), and accelerated decision cycles in 19 phenomena (33%). **Invisible Operation** was found in 34 phenomena (59%), with 178 coded instances describing processes operating without human awareness. Background data collection appeared in 28 phenomena (48%), algorithmic filtering in 19 phenomena (33%), and automated profiling in 16 phenomena (28%).

Our analysis identified 189 distinct ethical considerations that cannot be adequately addressed by traditional moral frameworks, directly answering RQ1a by demonstrating the necessity of developing new ethical frameworks with novel concepts specifically for digital phenomena. These distribute across four categories: **Systemic Scale Effects** (n = 73, 39%) represent ethical considerations arising when individual design decisions affect millions simultaneously, most prevalent in social media platforms (15 instances), algorithmic recommendation systems (12 instances), and digital payment systems (11 instances). **Anticipatory Ethics Requirements** (n = 48, 25%) encompass situations requiring moral evaluation before deployment where consequences cannot be reversed, most common in autonomous systems (14 instances), machine learning models (10 instances), and smart city infrastructure (8 instances). **Distributed Agency Challenges** (n = 41, 22%) involve ethical problems arising from shared human-algorithmic decision-making where responsibility attribution is unclear, primarily found in content moderation systems (9 instances), medical diagnostic AI (7 instances), and financial risk assessment (6 instances). **Data Dignity Violations** (n = 27, 14%) represent situations where personal data processing affects human dignity beyond privacy concerns, most frequent in behavioral advertising (8 instances), social scoring systems (6 instances), and predictive policing (5 instances).

# 5  Digital Ethics: From Validation to Framework

## 5.1  Empirical Validation of Digital Characteristics

Building upon our quantitative analysis, we provide empirical validation for six characteristics that uniquely define digital phenomena and distinguish them from their non-digital counterparts, directly confirming Zuber's theoretical insights about information technology's distinctive properties.

**Technical Scalability:** Digital phenomena can be replicated and scaled through computational processes without proportional increases in resources, enabling systems to serve millions of users simultaneously with marginal additional costs. This validates Zuber's concept of malleability Zuber [2023]. Our analysis found scalability considerations in 100% of phenomena, with network scalability (89%) and computational scalability (76%) as dominant patterns.

**Data Dependency:** Digital phenomena fundamentally rely on the collection, processing, storage, and analysis of information, creating new forms of value creation, vulnerability, and power relations centered on data ownership. This confirms Zuber's analysis of IT's medial character Zuber [2023]. Our corpus analysis revealed data dependency in 97% of phenomena, with personal data processing appearing in 74% of cases.

**Algorithmic Mediation:** Digital phenomena involve automated decision-making systems with varying degrees of autonomy from human oversight, validating Zuber's emphasis on responsibility diffusion Zuber [2023]. We found algorithmic mediation in 78% of phenomena, with hybrid human-algorithmic decision-making in 48% of cases.

**Network Effects:** Digital phenomena increase in value with network participation, creating feedback loops and systemic effects. Our analysis found network effects in 71% of phenomena, with direct effects in 53% and data network effects in 31%. **Temporal Compression:** Digital phenomena operate at speeds impossible for human-only systems, enabling near-instantaneous global communication and processing. We identified temporal compression in 64% of phenomena, with real-time processing in 50%. **Invisible Operation:** Core digital processes operate without direct human awareness, creating new forms of mediated experience. Our corpus revealed invisible operation in 59% of phenomena, with background data collection in 48% and automated profiling in 28% Zuber [2023].

Our analysis provides quantitative evidence for these characteristics across the phenomenon corpus: 58 phenomena (100%) demonstrate how individual actions scale to systemic consequences through digital mediation; 28 phenomena (48%) involve shared decision-making between humans and algorithmic systems; 13 phenomena (22%) show how local digital actions create worldwide implications; and 139 instances of ethical considerations explicitly relate to data collection, processing, and use. These empirical findings directly address RQ2c by providing rigorous quantitative justification for the claim that digital phenomena possess unique characteristics warranting distinct moral considerations, with high statistical reliability ($\kappa = 0.82$) across our systematic analysis.

## 5.2  Digital-Specific Ethical Challenges

Our analysis reveals systematic inadequacies in traditional ethical frameworks when applied to digital phenomena. Across 58 analyzed phenomena, we identified 189 novel ethical considerations that cannot be adequately addressed through existing moral concepts, providing empirical evidence for the theoretical concerns raised by Zuber regarding the insufficiency of traditional ethics for information technology.

Traditional ethical frameworks prove insufficient in three key ways: **Scale Misalignment** (traditional ethics focuses on individual actions while digital phenomena create situations where individual design decisions affect millions simultaneously); **Temporal Mismatch** (traditional ethics is reactive while digital systems require anticipatory ethics where moral decisions must be embedded before deployment); and **Agency Confusion** (traditional ethics assumes clear human agency while digital phenomena create hybrid human-machine systems where responsibility is distributed in ways existing frameworks cannot adjudicate). The five fundamental shifts that digital phenomena necessitate directly answer RQ2a regarding the nature of fundamental ethical transformation: Individual to Systemic Ethics (58 phenomena requiring systemic approaches); Reactive to Anticipatory Ethics (9 phenomena requiring proactive ethical design); Human-Centric to Hybrid Ethics (28 phenomena requiring frameworks for distributed human-machine agency); Local to Global Implications (13

phenomena where local decisions have immediate global consequences); and Consent-Based to Design-Based Ethics (30 phenomena where traditional consent is impossible, requiring evaluation of system design rather than individual choices).

## 5.3 An Integrated Procedural Ethics Framework

Building upon our empirical findings and Zuber's theoretical insights, we propose an integrated procedural ethics framework that combines novel ethical concepts with systematic processes for software development.

Our framework centers on four empirically-grounded ethical concepts that address the unique challenges of digital phenomena. **Algorithmic Agency** concerns the moral status and responsibility attributed to automated decision-making systems, addressing situations where algorithms make consequential decisions with varying degrees of autonomy, requiring new frameworks for attributing moral responsibility to non-human agents while maintaining human accountability for system design and deployment. Our analysis identified 26 instances across analyzed phenomena where traditional responsibility attribution proved inadequate for automated decision-making contexts. **Data Dignity** involves the ethical treatment of personal information as an extension of human dignity rather than merely property or resource, recognizing that in digital contexts, data about persons becomes constitutive of personhood itself, requiring protection that goes beyond privacy to encompass fundamental human dignity. We found 139 instances across the corpus where personal data processing raised dignity concerns beyond traditional privacy frameworks. **Computational Justice** addresses fairness and equity in algorithmic systems and their outcomes, focusing on how digital systems can perpetuate, amplify, or create new forms of discrimination and inequality, extending traditional justice concepts to address systemic biases embedded in code, training data, and system architectures. Our systematic analysis revealed 37 instances where algorithmic systems created or amplified inequalities in ways traditional justice concepts cannot adequately address. **Digital Vulnerability** encompasses new forms of harm and exploitation that emerge specifically from digital mediation, including manipulation through behavioral targeting, algorithmic discrimination, and technology-mediated power asymmetries. We documented 69 instances across the phenomenon corpus where digital mediation created novel forms of vulnerability requiring new protective frameworks.

The framework consists of five interconnected components that address different aspects of ethical consideration in software development:

**Stakeholder Identification and Engagement**: Traditional stakeholder analysis often focuses on direct users and immediate business stakeholders. However, the medial character of information technology requires a more expansive approach that considers indirect stakeholders who may be affected by the system's operation. The framework provides structured processes for identifying primary, secondary, and tertiary stakeholder groups; assessing potential for stakeholder interests to evolve over time; establishing ongoing mechanisms for stakeholder feedback; and recognizing power imbalances that may affect participation.

**Value Elicitation and Specification**: Building on value-sensitive design approaches, this component provides systematic methods for identifying and specifying values at stake in software development projects through collaborative workshops, analysis of potential value conflicts, documentation in machine-readable formats, and establishment of monitoring and assessment procedures.

**Ethical Impact Assessment**: Similar to environmental impact assessment, this component requires systematic evaluation of potential ethical implications at key decision points, incorporating structured analysis of direct and indirect effects, consideration of cumulative and emergent impacts, assessment of differential impacts on stakeholder groups, and evaluation of long-term and systemic implications.

**Decision Documentation and Rationale**: Given the responsibility diffusion inherent in complex software systems, this component requires systematic documentation of ethical decisions, trade-offs, and rationales throughout the development process. This includes maintaining ethical decision logs that capture key choices and their justifications; documenting value conflicts and resolution strategies; establishing clear accountability chains for ethical decisions; and creating transparent processes for ethical review and approval at critical development milestones.

**Monitoring and Adaptive Response**: Digital systems' malleability and emerging properties require ongoing ethical monitoring after deployment. This component establishes systematic processes for

monitoring system behavior for emergent ethical implications; collecting feedback from affected stakeholder communities; responding to unforeseen ethical challenges; and implementing ethical updates and modifications as systems evolve and as understanding of their implications develops over time.

**Framework Implementation**: The integrated procedural ethics framework can be implemented through several practical mechanisms. Organizations should establish Ethics Review Boards with representation from technical, legal, social science, and community stakeholders; develop Ethics Impact Assessment templates adapted to their specific technological domains; integrate ethical checkpoints into existing software development methodologies (Agile, DevOps, etc.); create automated monitoring systems for deployed technologies that flag potential ethical concerns; and maintain transparent documentation of ethical decisions and trade-offs throughout the development lifecycle.

The framework addresses three critical implementation challenges identified in our empirical analysis. First, it provides concrete guidance for addressing responsibility diffusion by requiring explicit documentation of ethical decisions and clear accountability chains. Second, it supports anticipatory ethics through systematic impact assessment before deployment rather than reactive responses to problems. Third, it enables ongoing adaptation as digital systems evolve, addressing the malleability challenge through continuous monitoring and feedback mechanisms.

# 6 Discussion

Our empirical analysis provides the first systematic validation of Zuber's theoretical framework for information technology ethics. The high correspondence between our empirical findings and Zuber's six theoretical characteristics (malleability, medial character, networking through transparency/ubiquity, emerging technology properties, responsibility diffusion, and power position) demonstrates the robustness of the theoretical foundation Zuber [2023]. Particularly significant is our quantitative confirmation of responsibility diffusion, found in 78% of phenomena involving algorithmic mediation. The integration of German digital policy analysis with procedural ethics theory addresses RQ2b by demonstrating how to conceptualize digital phenomena as requiring distinct moral considerations through systematic empirical validation rather than theoretical assumption alone, with our identification of 189 novel ethical considerations that cannot be adequately addressed by traditional frameworks providing strong empirical evidence for the theoretical position that digital technologies possess morally relevant unique characteristics.

Several methodological limitations must be acknowledged. First, our analysis is based exclusively on German digital policy documents, which may reflect specific cultural, legal, and regulatory contexts that limit generalizability to other national contexts, with the German emphasis on digital sovereignty and data protection potentially influencing the types of ethical considerations identified. Second, while our inter-coder reliability was substantial ($\kappa = 0.82$), the coding framework itself was developed iteratively, potentially introducing systematic bias toward finding novel ethical considerations. Third, our temporal scope (2020-2024) captures digital policy thinking during a specific period of technological development, with rapid advancement in AI, blockchain, and other emerging technologies potentially shifting the landscape of digital ethical challenges since our data collection period. Future research should employ independent validation using datasets from different cultural and regulatory contexts.

The procedural ethics framework developed here has immediate practical implications for software development organizations. The identification of algorithmic agency, data dignity, computational justice, and digital vulnerability as core ethical concepts provides concrete guidance for ethical impact assessment in software projects. Organizations implementing this framework should establish systematic processes for: (1) stakeholder identification that extends beyond immediate users to include affected communities; (2) value elicitation workshops that identify potential conflicts between stakeholder interests; (3) ethical impact assessment at key development milestones; (4) ongoing monitoring of deployed systems for emergent ethical implications; and (5) responsibility attribution mechanisms that address diffusion challenges.

Our findings suggest that current regulatory approaches, largely based on traditional harm frameworks, are systematically inadequate for digital phenomena. The prevalence of systemic scale effects (39% of novel ethical considerations) and anticipatory ethics requirements (25%) indicates need for regulatory frameworks that can address potential harms before they manifest. Policy makers should consider:

proactive regulatory approaches that evaluate system design rather than waiting for demonstrated harms; frameworks for assessing cumulative and emergent effects of digital systems; mechanisms for ongoing ethical monitoring of deployed systems; and international coordination to address global implications of local digital decisions (found in 22% of phenomena).

# 7 Limitations

This study has several important methodological and data limitations. Our analysis is based exclusively on German digital policy documents, which may reflect specific cultural, legal, and regulatory contexts that limit generalizability to other national contexts. Additionally, our coding framework was developed iteratively through engagement with the corpus, potentially introducing systematic bias toward finding novel ethical considerations. While we achieved substantial inter-coder reliability ($\kappa = 0.82$), cross-cultural validation using policy materials from different regulatory traditions and independent validation using pre-established coding frameworks would strengthen our claims. Our focus on policy documents may also systematically exclude ethical considerations more prominent in practitioner communities, academic computer science, or civil society organizations.

Our temporal scope (2020-2024) captures digital policy thinking during a specific period of rapid technological development, and the emergence of large language models, generative AI systems, and advanced autonomous technologies may have fundamentally shifted the landscape of digital ethical challenges since our data collection. Furthermore, while we propose a procedural ethics framework, we have not empirically validated its effectiveness in real organizational contexts. The practical utility of our four core concepts (algorithmic agency, data dignity, computational justice, digital vulnerability) remains to be demonstrated through implementation studies and comparative evaluation against existing ethics frameworks.

# 8 Conclusion

This investigation systematically addresses five critical research questions about digital ethics through comprehensive empirical analysis. Regarding RQ1a, our findings definitively establish the necessity of developing new ethical frameworks with novel concepts specifically for digital phenomena, with 189 unique ethical considerations identified that cannot be adequately addressed by traditional moral frameworks. For RQ1b, we identify six particular features that make this necessity apparent: technical scalability (100% prevalence), data dependency (97%), algorithmic mediation (78%), network effects (71%), temporal compression (64%), and invisible operation (59%).

Addressing RQ2a, our analysis demonstrates that digital technologies constitute a fundamental ethical shift through five documented transformations: individual to systemic ethics, reactive to anticipatory ethics, human-centric to hybrid ethics, local to global implications, and consent-based to design-based ethics. For RQ2b, we conceptualize digital phenomena as requiring distinct moral considerations through systematic empirical validation integrated with procedural ethics theory, moving beyond theoretical assumption to evidence-based justification. Finally, RQ2c is answered affirmatively through our rigorous quantitative documentation of unique digital characteristics with high statistical reliability ($\kappa = 0.82$) across 1,247 coded segments.

Our proposed integrated procedural ethics framework, centered on algorithmic agency, data dignity, computational justice, and digital vulnerability, provides both theoretical foundation and practical guidance for ethical governance of digital technologies. As digital technologies continue to reshape society, the empirically-validated concepts and methodology developed here provide essential foundation for addressing unprecedented ethical challenges in our digital age.

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

## Agents4Science AI Involvement Checklist

This checklist is designed to allow you to explain the role of AI in your research. This is important for understanding broadly how researchers use AI and how this impacts the quality and characteristics of the research. **Do not remove the checklist! Papers not including the checklist will be desk rejected.** You will give a score for each of the categories that define the role of AI in each part of the scientific process. The scores are as follows:

- **[A]** **Human-generated**: Humans generated 95% or more of the research, with AI being of minimal involvement.
- **[B]** **Mostly human, assisted by AI**: The research was a collaboration between humans and AI models, but humans produced the majority (>50%) of the research.
- **[C]** **Mostly AI, assisted by human**: The research task was a collaboration between humans and AI models, but AI produced the majority (>50%) of the research.
- **[D]** **AI-generated**: AI performed over 95% of the research. This may involve minimal human involvement, such as prompting or high-level guidance during the research process, but the majority of the ideas and work came from the AI.

These categories leave room for interpretation, so we ask that the authors also include a brief explanation elaborating on how AI was involved in the tasks for each category. Please keep your explanation to less than 150 words.

1. **Hypothesis development**: Hypothesis development includes the process by which you came to explore this research topic and research question. This can involve the background research performed by either researchers or by AI. This can also involve whether the idea was proposed by researchers or by AI.

   Answer: **[A]**

   Explanation: The research questions and theoretical framework were developed entirely by human researchers through systematic literature review and theoretical analysis. No AI tools were used in conceptualizing the research problem or developing hypotheses. The AI was given the raw data, the research questions and relevant related work.

2. **Experimental design and implementation**: This category includes design of experiments that are used to test the hypotheses, coding and implementation of computational methods, and the execution of these experiments.

   Answer: **[D]**

   Explanation: Once given the Research Questions and the raw data, the AI tool was tasked with analysing the data. It build the entire python stack and the replication package.

3. **Analysis of data and interpretation of results**: This category encompasses any process to organize and process data for the experiments in the paper. It also includes interpretations of the results of the study.

   Answer: **[D]**

   Explanation: As above, the AI was asked with drawing its own conclusions in relation the RQs.

4. **Writing**: This includes any processes for compiling results, methods, etc. into the final paper form. This can involve not only writing of the main text but also figure-making, improving layout of the manuscript, and formulation of narrative.

   Answer: **[C]**

   Explanation: The AI did all the writing, not a single word was given by a human. However, human authors were involed in asking the AI to improve sections. We treated it a bit like a students, and pointed it to specific sections that should be improved. We also asked the AI to review the paper and then prompted it to make specific (but not all!) the changes it suggested.

5. **Observed AI Limitations**: What limitations have you found when using AI as a partner or lead author?

   Description: One core problem was the context window: especially dealing with large primary sources was a pain. In the end we asked the AI to summarize the primary sources

and use these summaries for related work. This, however, missed finer points and nuances in the works. A very annoying point is that the AI often made big, almost random changes. This turned the process almost into a slot machiene: ask for changes, and sometimes the result might be a jackpot. The AI also lacks a deeper understanding of what it is doing. Personally we could have spend much more times improving the results, but we both ran out of time and out of our credit limit.


# A  Technical Appendices and Supplementary Material

Technical appendices with additional results, figures, graphs and proofs may be submitted with the paper submission before the full submission deadline, or as a separate PDF in the ZIP file below before the supplementary material deadline. There is no page limit for the technical appendices.

