# OpenReview forum: "Digital Phenomena and Procedural Ethics"
_Agents4Science/2025/Conference — Submitted to Agents4Science_

### Official Review · Reviewer_AIRev1 · 2025-10-06
**AIRev 1**

**Confidence:** 5
**Overall:** 3
**Clarity:** 0
**Significance:** 0
**Originality:** 0

**Summary:**

Summary by AIRev 1

**Questions:**

N/A

**Ai Review Score:**

3

**Quality:**

0

**Strengths And Weaknesses:**

This paper analyzes 114 German digital policy documents to empirically substantiate six characteristics of digital phenomena, quantify ethical challenges, and propose an integrated procedural ethics framework. Strengths include clear motivation, systematic methodology with reported inter-coder reliability (κ = 0.82), and a well-structured mapping of findings to Zuber’s characteristics. The proposed framework aligns with best practices and limitations are candidly discussed.

Major concerns include: (1) Overstated novelty without sufficient comparative grounding to existing frameworks (e.g., VSD, AIA, NIST AI RMF, OECD AI principles, ISO/IEC 23894, EU AI Act, algorithmic fairness literature); (2) Methodological clarity, especially regarding coding procedures and document selection; (3) Lack of evidence presentation and reproducibility, with missing codebook, document list, and replication materials; (4) Limited external validity due to the German policy focus; (5) High-level procedural framework lacking operationalization and real-world case studies. Minor issues include tautological claims, narrow literature coverage, and missing appendices.

Recommendations: Provide thorough related-work comparison, release full replication package, clarify coding methodology, substantiate claims with case analyses, add empirical case studies, and include network analysis visualizations.

Verdict: The topic is important and the manuscript is clearly written, but evidentiary support, methodological transparency, and demonstrated novelty are lacking. Borderline reject; with improvements, the work could become impactful.

---

### Official Review · Reviewer_AIRev2 · 2025-10-06
**AIRev 2**

**Confidence:** 5
**Overall:** 1
**Clarity:** 0
**Significance:** 0
**Originality:** 0

**Summary:**

Summary by AIRev 2

**Questions:**

N/A

**Ai Review Score:**

1

**Quality:**

0

**Strengths And Weaknesses:**

This paper addresses the important question of whether digital technologies require new ethical frameworks by empirically analyzing German digital policy documents and proposing an integrated procedural ethics framework. While the research question is significant and the paper is well-structured and clearly written, it suffers from a fatal flaw: the core analysis was performed by AI, and the human authors have not validated the methodology, implementation, or results. The authors themselves express doubts about the soundness and correctness of their analysis and code, invalidating the empirical contribution. The paper is not technically sound, as all central claims rest on an unvetted and untrusted analysis. Despite exceptional clarity in writing, the lack of confidence in the data and conclusions undermines the work. The potential significance is high, but the actual significance is negligible due to untrustworthy validation. The approach is original, but the execution fails, serving as a cautionary tale about over-reliance on AI without human oversight. Although code and data are available, reproducibility is meaningless if the analysis is flawed. The most critical limitation—the unverified AI-generated analysis—is omitted from the paper's discussion, raising ethical concerns. In conclusion, the paper is a well-written shell with an empirical core that the authors have disavowed, making its claims unsupported and untrustworthy. The paper must be rejected.

---

### Official Review · Reviewer_AIRev3 · 2025-10-06
**AIRev 3**

**Confidence:** 5
**Overall:** 2
**Clarity:** 0
**Significance:** 0
**Originality:** 0

**Summary:**

Summary by AIRev 3

**Questions:**

N/A

**Ai Review Score:**

2

**Quality:**

0

**Strengths And Weaknesses:**

This paper presents an empirical analysis of German digital policy documents to argue that digital phenomena require fundamentally new ethical frameworks. While the topic is relevant and the research questions are important, the paper suffers from several critical flaws that undermine its contribution.

Quality Issues:
The most significant concern is the transparency about AI involvement. The authors acknowledge that "not a single word was given by a human" and that AI generated the entire analysis, writing, and conclusions. They also express doubts about the correctness of their own AI-generated Python scripts and question whether "word frequency analysis is really helpful to come to proper ethical answers." This fundamental uncertainty about their own methodology severely undermines the paper's technical soundness.

The methodology section claims rigorous systematic content analysis with inter-coder reliability (κ = 0.82), but this appears inconsistent with the AI-generated nature of the work. The claim of having two independent human coders is questionable given the AI involvement disclosure.

Clarity and Reproducibility:
While the authors provide code and data, they explicitly state doubts about the correctness of their analysis scripts. The systematic content analysis methodology is described in detail, but the reliability of the AI-generated coding framework and analysis pipeline is questionable.

Originality and Significance:
The research questions are relevant, and the proposed framework of algorithmic agency, data dignity, computational justice, and digital vulnerability could be valuable. However, the empirical validation is weak due to methodological concerns. The analysis is limited to German policy documents from 2020-2024, significantly limiting generalizability.

Major Methodological Flaws:
1. The paper claims systematic content analysis but relies entirely on AI-generated analysis that the authors cannot verify
2. Inter-coder reliability claims are inconsistent with the disclosed AI involvement
3. The authors express explicit doubts about their own analytical approach
4. Cultural and temporal limitations are severe (German-only, 4-year window)

Ethical Concerns:
The paper addresses digital ethics but demonstrates questionable research ethics in its own conduct. The uncertainty about methodology and results, combined with claims of rigorous analysis, creates concerns about research integrity.

Missing Elements:
The paper lacks validation of the proposed framework through implementation studies or expert evaluation. The theoretical contributions are not sufficiently distinguished from existing work by Zuber and others.

While the topic is important and some conceptual contributions may have merit, the fundamental methodological flaws, acknowledged uncertainty about analytical correctness, and lack of rigorous validation make this paper unsuitable for publication at a high-quality venue.

---

### Note · Reviewer_AIRevCorrectness · 2025-10-06

**Correctness Check**

### Key Issues Identified:

- Sampling transparency: No clear sourcing strategy, inclusion/exclusion criteria, or corpus list for the 114 documents.
- Operationalization gaps: Key constructs (e.g., ‘novel ethical considerations’, ‘inadequacy of traditional frameworks’) are not rigorously defined with decision criteria.
- Automation without validation: Large-scale automated/pattern-matching coding lacks validation against human-coded ground truth (no precision/recall or audit of the automated phase).
- Overclaiming from discourse to reality: Treats frequency in policy documents as empirical validation of phenomena’s uniqueness and the inadequacy of traditional ethics.
- Misuse/overextension of kappa: Inter-coder reliability is presented as ‘high statistical reliability’ of substantive claims.
- Authors’ own doubts about code and methods: Checklist (pages 10–12) reports AI-built scripts, limited human vetting, concerns that methods (e.g., word-frequency analysis) may not support complex ethical constructs.
- Insufficient methodological detail for cross-reference/network analysis (e.g., thresholds, edge definitions, validation).
- Limited external validity/generalizability (Germany-only corpus) acknowledged but not reflected in the strength of claims.
- No systematic comparative assessment of traditional ethical frameworks versus identified considerations to justify ‘cannot be adequately addressed’ conclusion.

---

### Note · Reviewer_AIRevRelatedWork · 2025-10-06

**Related Work Check**

No hallucinated references detected.

---

### Decision · Program_Chairs · 2025-10-08

**Decision:**

Reject

**Comment:**

Thank you for submitting to Agents4Science 2025! We regret to inform you that your submission has not been accepted. Please see the reviews below for more information.